# Outcomes of a Teaching Learning Sequence on Modelling Surface Phenomena in Liquids

Onofrio Rosario Battaglia *, Aurelio Agliolo Gallitto, Giulia Termini and Claudio Fazio *

Department of Physics and Chemistry-Emilio Segrè, University of Palermo, 90128 Palermo, Italy
* Correspondence: onofriorosario.battaglia@unipa.it (O.R.B.); claudio.fazio@unipa.it (C.F.)

**Abstract:** In this paper we discuss the effects of modelling and computer simulation activities in promoting student use of lines of reasoning useful to explain proposed or observed situations. The activities are part of a structured Teaching/Learning Sequence on surface phenomena in liquids. We outline a model of liquid based on a mesoscopic approach, examples of computer simulations students can use during the activities, and we describe the Teaching/Learning Sequence. During the pedagogical activities, students can simulate the liquid behaviour by controlling many simulation parameters, such as the interaction intensity among liquid and solid particles. The results of the analysis of student answers to a questionnaire before and after instruction, and of other qualitative data, show that these activities can help the students to think in terms of "mechanisms of functioning".

**Keywords:** surface phenomena; surface tension; mesoscopic model; active learning; educational reconstruction; SPH methods

## 1. Introduction

Model-based methods of physics instruction and learning have gained popularity in the literature over the past few decades (e.g., [1]). According to several studies, modelling is a superordinate process skill (e.g., [2]). Other studies indicate that the inclusion of modelling activities in physics classes can help foster a positive attitude toward learning physics, improve students' comprehension of physics [3], and enhance students' epistemological beliefs [4] about models and their use in learning science [5]. Moreover, a teaching approach focused on modelling can improve students' reasoning, supporting them in seeing similarities among a wide range of phenomena [6,7] that, at a first analysis, may appear different.

An effective way to introduce students to modelling is to engage them in collaboratively using interactive simulations that allow them to actively modify some parameters of the models and to see and discuss in real-time the effects, as reported by Rutten et al. [8,9]. Students are engaged in asking questions, collecting, discussing and evaluating data to evaluate and find answers to questions that are posed by the teachers or which arise from the discussion of the results, in a framework that is typical of inquiry-based learning (e.g., [10]).

In this article, we discuss pertinent modelling aspects that are part of a Teaching/Learning Sequence (TLS) (e.g., [11,12]) on liquid surface phenomena that emphasizes the participation of secondary school students in inquiry- and investigative-based activities. In particular, we focus on the advantages offered by modelling and interactive computer simulation activities in promoting student use of lines of reasoning [13] useful to explain proposed or observed situations. The choice of the surface phenomena as a topic was driven by the consideration that the foundations and applications of this topic are relevant to many scientific and technical fields, such as physics, engineering, medicine, and environmental sciences. Understanding surface phenomena, such as capillarity and drop formation, involves thermodynamics, statistical mechanics, fluid mechanics, and other science subjects.

However, teaching these topics at secondary level is often not performed during physics lessons. Surface phenomena are generally taught as a small part of chemistry courses. These phenomena are very often discussed only macroscopically by introducing a quantity called surface tension; sometimes by thermodynamical approach, other times by introducing a force per unit length, still other times by cohesive and adhesive forces. The thermodynamical approach can provide an adequate macroscopic description of surface phenomena, but it often turns out to be too abstract for secondary school students. The approach based on forces is sometimes incoherent and often lacks deep insight into the nature of the forces. Thus, surface phenomena are often perceived by students, and even teachers, as obscure and not so relevant for educative purposes, even at the university level [13–15].

A recent paper introduces a new and quite complete explanation of surface tension in macroscopic terms [16]. However, the approach proposed can be conceptually complex for secondary school or even undergraduate students.

An alternative (and more complex) way to describe and explain surface tension involves microscopic modelling, which takes into account molecular interactions and thermal effects. A microscopic model for surface phenomena is, in principle, suitable from a didactic point of view, as it has the advantage to clearly highlight the interactions among the particles responsible for these phenomena, and it can be implemented in computer-based simulations. However, it cannot be easily used at school, or even the undergraduate level, especially when teachers want to use computer-based simulations that aim to describe the behaviour of large portions of liquid. Relevant computing resources, which are not always available, would be necessary, especially at secondary school level.

In the last years there has been some interest in the physics education literature about the use of "mesoscopic" models [17] to describe friction or liquid behaviour in terms of interactions among "particles". Mesoscopic models have also been used to introduce students to the study of surface phenomena in liquids by means of interactive simulations [18,19]. In those approaches, the liquid is considered as composed of particles with a radius equal to a tenth or hundredth of a millimetre. The size of these particles depends on the spatial resolution and the calculation efficiency, and the interaction forces between particles are similar to those present at the microscopic level. The user of the simulations can experience the advantages of the microscopic level and can also simulate large portions of liquids readily available in schools, even with low-cost computers.

Several simulation results of liquid behaviour based on mesoscopic models are described in the literature, often in complex experimental situations and in the engineering field [20,21]. Many of them are implemented by means of Smoothed-Particle Hydrodynamics (SPH) computational methods [22]. However, very few examples exist of basic phenomena that can be used at an educational level, such as the formation of a liquid drop or the formation of menisci [18,19].

Here, after a discussion on the main aspects of a mesoscopic model of liquids, we present two relevant examples of pedagogical computer-based simulations we implemented by using the model, and we discuss the main aspects of the TLS structure. Finally, we present some results of the trialling of the TLS with respect to the efficacy of this approach in fostering student reasoning lines oriented to explaining situations and phenomena.

## 2. The Mesoscopic Model of Surface Tension

In our model, a generic liquid is made of mesoscopic particles. The equations of motion (Navier-Stokes equations) of the discretized liquid are solved through the SPH computational method.

SPH is a Lagrangian numerical method used to obtain approximated solutions of the equations of fluid dynamics, modelling the liquid by a set of mesoscopic particles. In this mesoscopic model, the liquid is composed of particles whose size is much bigger than that of a molecule (commonly, particle radius ranges from 0.1 mm to 1 cm). SPH particles are in fact large portions of liquid each containing a huge number of molecules. This choice

makes the SPH approach capable of simulating even very large portions of liquid without the need of large computational resources and without loss of detail.

In the SPH method, liquid physical properties (mass, density, etc.) are associated with each particle. Each physical quantity is obtained by interpolation among many particles through specific functions called "kernel" functions. A typical and fundamental parameter in the SPH method is the dominion (or support) of these functions, usually called "smoothing length". The smoothing length significantly affects the simulation efficiency and accuracy. This value must be large enough to cover a large enough number of particles with its domain. However, it must be smaller than the typical dimensions of the system to simulate, in order not to lose resolution and efficiency.

For the dynamic study of our system, we took into account the "pressure" force, the viscosity force and the gravitational force.

It is worth remembering that pressure is a macroscopic quantity and certainly cannot be defined at a microscopic level. In the SPH approach, the liquid is composed of particles as small as millimetres or tens of millimetres in size. These particles are actually portions of liquid and it is therefore possible to consider pressure on a particle.

To simulate surface tension phenomena, it is necessary to add in the model a further additive force $F_{ij}$ between two generic particles $i$ and $j$. Since the origin of surface tension is strictly connected to inter-molecular interactions, it is reasonable that this interaction acting between mesoscopic particles has the same properties as the forces between microscopic particles. They are attractive at large distances and repulsive at short distances. The value of this force depends on the two interacting particles (liquid-liquid or solid-liquid) and on the inter-particle distance.

According to Tartakovsky and Meakin [23], the $F_{ij}$ force, called here "molecular-like" force, can be defined by using the function

$$F_{ij} = \begin{cases} S_{ij} cos\left(\frac{3\pi}{4H}|r_j - r_i|\right) & |r_j - r_i| \leq 2H \\ 0 & |r_j - r_i| > 2H \end{cases} \tag{1}$$

where $S_{ij}$ and $|r_j - r_i|$ indicate the magnitude of the force between two particles $i$ and $j$ and the distance between them, respectively.

To limit the amount of computed inter-particle interactions and achieve good computational efficiency, it is necessary to set a cut-off on the interaction range. We decided to set the cut-off at a distance equal to $2H$, unlike Tartakovsky and Meakin [23] where the analogous mathematical function has a support equal to $H$.

The force $F_{ij}$ is antisymmetric (i.e., $F_{ij} = -F_{ji}$) as requested for momentum conservation. The interaction force is repulsive when the inter-particle distance is lower than $\frac{2}{3}H$, attractive when the inter-particles distance is in the range $\left[\frac{2}{3}H, 2H\right]$, and zero when the inter-particle distance exceeds $2H$. A graphical representation of this force is shown in Figure 1.

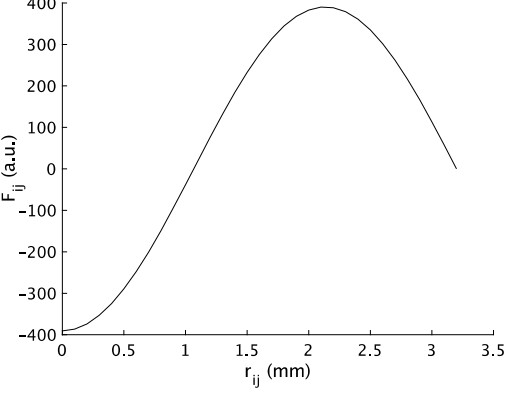

**Figure 1.** The module of the molecular-like force as a function of the inter-particle distance $r_{ij} = \left|r_j - r_i\right|$ between a particle $i$ and a particle $j$. $H$ = 0.16 mm.

It is worth noting the fundamental role of the short-range repulsive force for obtaining liquid behaviour coherent with experimental evidence, as reported in many research results [18,24,25].

This force allowed us to describe many surface phenomena, such as capillary phenomena, by modelling the interactions between liquid particles or liquid particles and solid particles. [18].

### 3. Some Examples of Simulation Activities

The model described above was implemented by a custom-built Fortran code. All the simulation results reported in this section were obtained by running this code on common personal computers. MATLAB software was used for graphs and movies.

### 3.1. Menisci Formation

The behaviour of a liquid in a vessel is simulated by starting from an arrangement of SPH particles with gravity and not in mechanical equilibrium. The vessel is made of fixed SPH particles. By varying the values of the $S_{ij}$ intensity of the molecular-like force, we qualitatively studied the formation of menisci [19].

When the intensity of the interaction $S_{ij}$ between two generic liquid SPH particles were close to the intensity between a liquid particle and a solid one, we obtained at the equilibrium—after several time steps—a result similar to the one shown in Figure 2. This case is comparable to what happens for liquids like water in a glass vessel where the concave shape of the meniscus is experimentally observed.

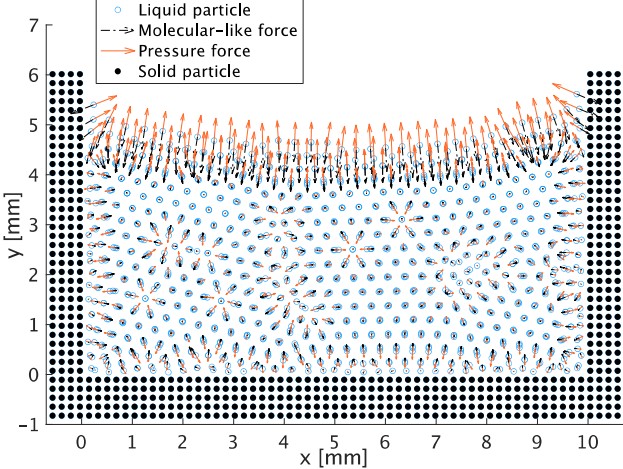

**Figure 2.** Liquid particles in a tank at equilibrium with initial inter-particle distance equal to 0.2 mm for water-like liquid. The SPH particles making the vessel (black dots) are fixed. Black dashed arrows are the resultant of the molecular-like forces, and the red ones are the pressure forces.

When the intensity $S_{ij}$ between two generic liquid SPH particles was remarkably greater than the interaction between a liquid particle and a solid one we obtained a result like the one shown in Figure 3. This case is comparable to what happens for liquids like mercury in a glass vessel, where the convex shape of the meniscus is experimentally observed.

### 3.2. The Pressure inside a Droplet

At the beginning of the simulation, SPH particles were homogeneously arranged with inter-particle distance $d_s$ into a rectangular configuration without gravity and without being in mechanical equilibrium. The simulation results were obtained by setting the liquid–liquid interaction force $S_{ij} = -10^{-5}$ a.u. and the initial inter-particle distance equal to 0.16 mm.

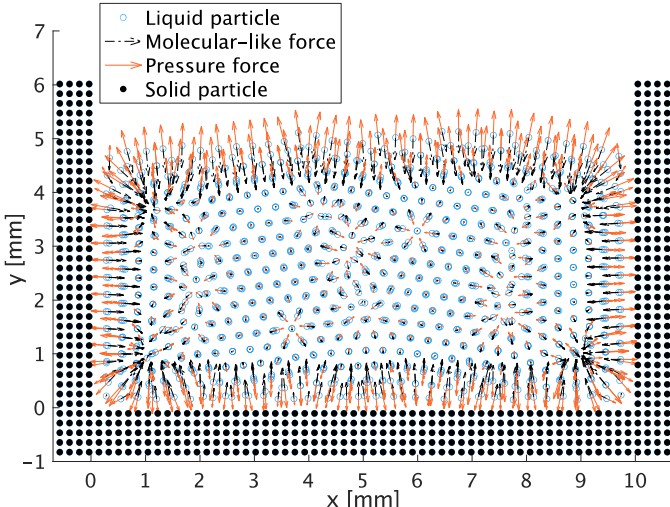

**Figure 3.** Liquid particles in a tank at equilibrium with initial inter-particle distance equal to 0.2 mm for mercury-like liquid. The SPH particles making the vessel (black dots) are fixed. Black dashed arrows are the resultant of the molecular-like forces, and the red ones are the pressure forces.

We performed many simulations to obtain the formation of bidimensional drops of a given liquid for different radii. For each simulation (i.e., for given dimensions of the initial rectangle) we obtained the radius of the droplet at the equilibrium, after many time steps. Figure 4 shows an example of a droplet at the mechanical equilibrium where the molecular-like and the pressure forces are also highlighted.

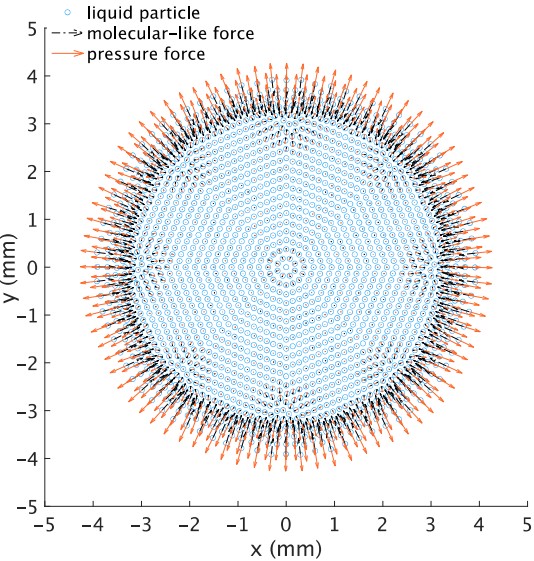

**Figure 4.** An example of a droplet at the equilibrium obtained after several simulation steps.

At the mechanical equilibrium, the pressure inside a liquid droplet depends on the surface tension and droplet radius according to the following equation

$$P_T = \frac{\sigma}{R} \tag{2}$$

called the Young-Laplace law for the bidimensional case, where $\sigma$ is the surface tension, $R$ the radius, and $P_T$ the pressure inside the drop. The surface tension can be determined by taking into account Equation (2) once the pressure $P_T$ inside the droplet is known.

Hoover [26] has shown that the SPH representation of a liquid is isomorphic with molecular dynamics, with many-body particle-particle interactions. This allows the SPH equations and the particle-particle interactions to be treated in a consistent manner.

By taking into account the Virial Theorem [27], the pressure $P_T$ can be calculated from the particle-particle interaction force as

$$P_T = P_k + \frac{1}{4\pi r^2} \sum_i \sum_j \boldsymbol{r}_{ij} \cdot R_{ij} \quad i \neq j \tag{3}$$

where $P_k$ is the ideal gas (kinetic) contribution to the pressure, $r$ is a radius of a circle inside the liquid drop, and $R_{ij}$ is the particle-particle interaction force. The summation in Equation (3) is over all the $i$ particles that lie inside the radius $r$ and all the $j$ particles in the drop. Self-interactions are excluded. It is worth noting that our simulation works in a bidimensional space and for this reason we calculated the pressure as the ratio between the "virial" and the area $4\pi r^2$ of a circle of radius $r$ inside the droplet (see Equation (3)).

At mechanical equilibrium the viscous forces are zero. Therefore, the interaction force is

$$R_{ij} = T_{ij} + \mathrm{F}_{ij.} \tag{4}$$

where $T_{ij}$ is the "pressure" force. By taking into account Equations (3) and (4) and considering that, at the equilibrium $P_k = 0$, we obtained as one of the outputs of the simulation the pressure inside the droplet.

Figure 5 shows the pressure $P_T$ calculated by using Equation (3) as a function of the inverse of the radius $R$ and a linear fit by using Equation (2), for a given simulated liquid, by varying the droplet radius.

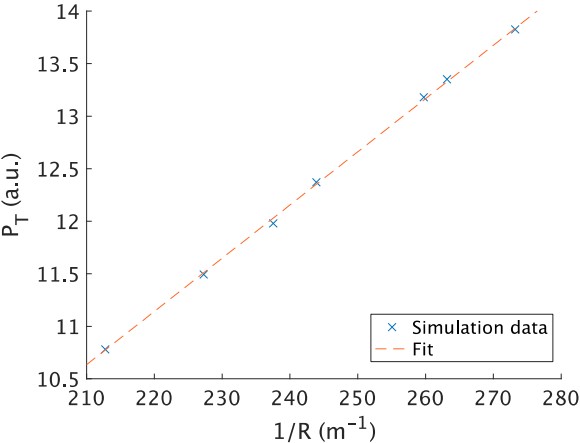

**Figure 5.** Droplet pressure as a function of the inverse of the droplet radius for a given liquid and linear fitting. The Determination Coefficient is about 0.99.

The linear regression line reported in Figure 5 clearly shows a good agreement between the simulations results and the Young-Laplace law.

## 4. The Student Sample and the Teaching/Learning Sequence

### 4.1. The Student Sample

The sample on which we trialled the TLS was made up of 19 students attending the fourth year of an Italian science-oriented Upper Secondary School ("Liceo Scientifico"). This sample was randomly selected from a group of 38 students from the same school and was proposed with a mesoscopic approach to the study of surface tension. The other 19 students were proposed with a traditional macroscopic approach to the study of surface phenomena. The results of the trialling with this second student group are discussed elsewhere [28].

*4.2. The TLS*

The TLS is based on active learning activities founded on observation, experiments (real and simulated), modelling activities, small and large group discussions, etc. It was pilot trialled during the 2021–22 school year with Upper Secondary School students. The activities took place over the course of 24 h, which were divided into 6 working afternoons.

One of the aims of the research was to study the impact of a modelling and simulation approach to surface phenomena on students' understanding, with particular attention to the lines of reasoning deployed by the students when trying to make sense of proposed or observed experimental situations.

The theoretical framework and the methodological approach on which the TLS was based refer to the model of Educational Reconstruction (e.g., [29]), and to the general idea of active learning (e.g., [30]), respectively.

As it is well-known, according to the Educational Reconstruction model, the physical content to be taught must not be simply taken from textbooks as it is and presented to students in an uncontextualized form. It must be reconstructed, adapting the physical content to the students, the learning knots known from the research, and to the specific learning environment and context. Moreover, according to the active learning-based approach to science education, to generate interest in learning students must be authentically involved in the pedagogical activities and, more specifically, in science practices miming scientific research. In particular, they must alternate work in small and large groups; when working in small groups, they are engaged in observing natural phenomena, posing questions related to what they observe, formulating answers and multiple explanations for them, and systematically testing them. When discussing in large groups, they cooperate and share their ideas, continuously review and improve the models built in small groups, and compare and contrast them within a model of scientific community created within the class.

Special attention was paid during the planning and implementation of the TLS to the way each student works in the team he/she belongs to and approaches physical situations and their description/explanation. Students were encouraged to not only cooperate with their teammates, but also to personally reflect on what is done. In this way students could independently search sources and material that helped them in carrying out the activities and consolidating the acquired knowledge, and were therefore able to proactively discuss the results with the team mates. During the small and large group activities, the students were actively involved in planning, doing observations and experiments, and constructing descriptive and explanatory models of the phenomena observed. Particular attention was paid by the teacher to activating gradually increasing levels of complexity and difficulty for the tasks proposed (e.g., [31].). Moreover, special care was given in our approach to propose activities that could fit the possible different learning and cognitive styles the students carry (e.g., [32,33]), to optimize the perception, gathering and processing the learning materials [34]. At the end of the activities, we invited each team to produce a report on their scientific experience, using notes, photos, videos and everything they had collected and recorded.

During the first lesson of the TLS, students were given a pre-instruction questionnaire on basic issues connected to surface phenomena but unrelated to the topics presented and analyzed during the course. The same day, after a thirty-minute break, the students were given a second questionnaire containing questions about the teaching/learning sequence. Both the questionnaires have been designed and validated according to methods well-known in the literature [35]. Validation of the questionnaires was conducted on samples other than those used to trial the teaching/learning sequences. These samples consisted of fourth-year students from the same school as the research sample.

The observation of phenomena by means of audio/video material, gathering of further information through different media (books, journals, web videos, websites, etc.) was performed in the classroom, at increasing levels of complexity and involvement of the students. The experimental modelling and simulation activities were performed with progressively increasing levels of complexity too, and information was conveyed and shared by using

several communication styles. During the activities, students were first asked to make previsions and to compare them with the experimental results; then they were individually invited to discuss and express their agreement with the group conclusion. Students were always requested to reflect on their perceptions of self-efficacy in understanding [36,37] and their well-being in learning [38].

All the students participated in qualitative and quantitative laboratory activities. Qualitative laboratory activities were carried out according to a bounded inquiry approach [39] in which students were free to choose which tools to use and which path to follow to answer the questions asked by the teachers. Quantitative activities were carried out according to a guided inquiry approach [39–41] in which students were guided during the activities that led them to answer the questions asked by the teachers.

We designed quantitative experiments to study surface phenomena and to give estimations of surface tension values in different liquids, in addition to qualitative experiments useful for introducing the students to the situations to be analyzed. In some cases, starting from well-known experimental set-ups, we reconstructed them with a very low budget, using materials available in ordinary didactic laboratories. Through the well-known Du Nouy ring method—redesigned with the aim of using easily available and cheap material—students were able to measure surface tension values for different liquids, such as commercial demineralized water, 99% ethyl alcohol, pure glycerol, common commercial peanut, sunflower, corn oil, and 30 mL dishwashing soap with demineralized water mixture. The ring was made of aluminium, and the forces involved were measured using a digital balance (sensitivity: $10^{-4}$ N). In another experiment, students were able to estimate the surface tension of soap bubbles as the radius of the bubbles varied so as to verify the Young-Laplace law (For more detail, see [42]). Finally, in a third experiment, students were able to estimate the contact angle between a liquid such as water and glass.

The modelling and simulation activities of the TLS were based on the mesoscopic model of liquids discussed in Section 2. Students were initially introduced to the model without discussing the mathematical details of the SPH method. Students were only required to understand the types of interactions between particles by discussing the pressure force and the molecular-like force whose modulus is represented in Figure 1. Particularly, they focused on the different roles played by forces over small and large distances, and on different interactions between two "liquid" particles, and between "solid" and "liquid" particles.

By using numeric simulations based on the SPH method, students were able to control relevant model parameters (for instance, the interaction intensity $S_{ij}$ in Equation (2)) and compare the results of the simulations with the experimental results. Students were encouraged to use the computer tools to manipulate the main quantities of the models and to visualize the simulation results. They actively discussed and compared the results collected by the various groups.

By using the simulations, students were able to study different surface phenomena, mainly qualitatively and, in some cases, quantitatively. Qualitatively, they could correctly reproduce the formation of a liquid drop with and without gravity (like a droplet coming out of a tap), the formation of menisci in a vessel, the sessile drop and a solid resting on a liquid. The primary objective was to enable students to comprehend, at least qualitatively, that the sources of these phenomena are particle interactions, [19] by visualizing (and making movies by means of the simulation software) the main forces acting on each particle, as shown in Figures 2–4. The students could modify the interaction intensity $S_{ij}$ and observe the effects of these changes both on the mechanical equilibrium reached by the studied system and on the temporal evolution. Quantitatively, they could study the rise of a liquid in a capillary tube and the Young-Laplace law. For example, the students could modify the values of interaction intensity $S_{ij}$ and gravitational acceleration, and observe how these changes affected the liquid level reached along the capillary tube and meniscus curvature radii which are formed inside the capillary tube and outside in the vessel. In this case, students could also estimate the capillary length for this system as $l_c = \left( \frac{\sigma}{\rho g} \right)^{1/2}$ and

compare it with the curvature radius, once the surface tension $\sigma$ is found and the density $\rho$ and gravity acceleration $g$ are given. The effect of gravity can be negligible compared to the effect produced by surface tension when the curvature radius is smaller than the capillary length. By simulating the formation of the drop of liquid in the absence of gravity as the radius of the drop varies, they could observe and "measure" how the pressure changes inside the drop and obtain an estimate of the surface tension value for a given liquid through the Young-Laplace law.

At the end of the activities, students answered two post-instruction questionnaires identical to the pre-instruction ones. A satisfaction questionnaire on the TLS activities and methodologies was also administered to the students after instruction. A couple of months after the end of the activities, students answered a questionnaire identical to the post-instruction one, dealing with topics specific to the teaching/learning sequences.

All the answers to the questionnaires and the observations made during the development of the trial were used to understand what aspects of the teaching/learning sequence could be considered most effective in fostering an effective and long-standing appropriation of the topics; the development of reasoning skills oriented to explanation rather than to a mere description of what was observed/done; and the development of positive attitudes towards the study of science.

## 5. The Results

One of the goals of this study is to demonstrate the effectiveness of the modelling approach in developing student reasoning lines oriented to explanations of events and phenomena.

Here, we discuss some results of the analysis of the students' responses to the pre/post-instructions questionnaire made of 11 open-ended questions dealing with topics specific to the teaching/learning sequence. The questionnaire, and the typical responses given by the students to the questions (e.g., [6]), can be found in the Appendix A. The aim is to highlight how, before instruction, students describe and explain surface phenomena in situations related to everyday life contexts; and if, and how, these descriptions/explanations are modified by the TLS attendance.

The analysis of students' answers to the questionnaire was initially conducted via phenomenographic methods [43], and then refined by using content analysis [44]. It allowed us to identify, with the basis of previous research [7], three students' "epistemological profiles" related to three different ways to reason when tackling the situations proposed in the questions. The profiles are shown in Table 1, where a brief description of the reasoning procedures that the students used when tackling the questions is given for each profile.

**Table 1.** Students' "epistemological profiles" related to the ways to tackle the questionnaire and the related reasoning procedures.

| Practical/Everyday | Descriptive | Explicative |
|---|---|---|
| Reflects the creation of situational meanings derived from everyday contexts. The student uses other situations, perceived as analogous to the one proposed in the question, to try to describe/explain it. | The student describes and characterizes the proposed situation/analyzed process by searching in memory the variables perceived as relevant and/or recalling their relations. The variables and the relationships among them are expressed by means of different languages (verbal, iconic, mathematic). Causal relations among the variables on the basis of a functioning model (microscopic/macroscopic) are not given. | The student gives an explanation of the proposed situation referring to a model (qualitative and/or quantitative) based on cause/effect relations. He may also provide explanatory hypotheses by introducing models which can be seen at a theoretical level. |

Table 2 shows examples of key words and sentences that, on the basis of the content analysis of students' answers, allowed us to classify the answers in one of the three abovementioned profiles.

**Table 2.** Examples of terms and sentences in students' answers used to classify them in one of the three "epistemological profiles" described in Table 1.

| Practical/Everyday | Descriptive | Explicative |
|---|---|---|
| (according to my) experience . . . Like I see in real life . . . usually . . . real object . . . like an insect on water . . . | I remember that . . . I studied that . . . I know that . . . The formula says . . . There are adhesive and cohesive forces . . . There is surface tension . . . Chemistry/Physics says . . . | molecular movement . . . . is similar to . . . microscopic . . . inter-molecular forces . . . interaction . . . equilibrium . . . molecules . . . |

The bar diagrams in Figure 6 show the results of the analysis for our sample of students. They show that, before instruction, 43% of the possible answers were not given, and the given answers were mainly the everyday- or descriptive-type. After instruction, the number of not-given answers drastically reduced to 1% of the total. Everyday-type answers are absent, and a relevant increase in explicative-type answers is observed. An increase in descriptive-type answers is also observed.

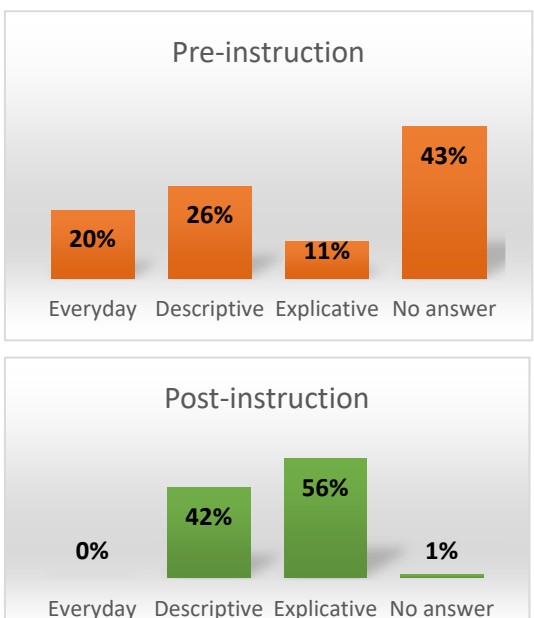

**Figure 6.** Bar diagrams showing the answers given by the students to the pre- and post-instruction questionnaire. The answers are categorized according to the "epistemological profiles" identified in our analysis and reported in Table 1.

A more detailed analysis of the answers given to the questionnaire is in progress, as well as the analysis of the answers to the other questionnaire related to topics not specific to the TLS.

Data coming from other evaluation tools—such as student worksheets and final scientific reports, logbooks, audio recordings of discussions during the experimental activities and interviews—are also currently being studied in order to get more information on other relevant study variables related to the promotion of student learning due to the TLS activities [28].

A preliminary analysis of interviews with students and the scientific reports they prepared after the activities allow us to give more detail on the development of student reasoning skills more oriented to explaining rather than describing. For example, at the end of one of the simulation sessions one of the students commented on his experience during the session: "Before taking part in these activities, I was sure that adhesive and cohesive forces were the causes of surface tension, because I studied that in Chemistry. However, I never reflected on what causes these forces and where they are applied. The simulation allowed me to understand that interactions among particles are the cause of the forces. The possibility to actually see the forces acting on the particles and their effects was also important".

Here, the student demonstrates that the simulation enabled him to shift from a descriptive approach—based on the uncontextualized recall from memory of content studied in chemistry—to an approach more focused on a search for a cause-and-effect mechanism, and promoted a reflection on the origin of forces and their application points. Furthermore, highlighted is the importance of the student being able to visualize the forces.

Another student said: "The simulation allowed me to make sense of the formation of a water drop and of the spherical shape it has (in absence of gravity). I enjoyed the possibility to see the forces acting on the particles at the drop surface and the modifications on the forces due to varying some simulation parameters. That made me easier to understand the reason of the spherical shape".

Here, the student expresses appreciation for the chance to explain the drop's spherical shape in terms of forces acting on the particles at the drop's surface (see Figure 4). She further notes that the ability to adjust simulation parameters and observe the effects on forces helped her comprehend the equilibrium shape of the drop.

Generally, we found that, after the simulation-based instruction, students were more able than before to explain the observed situations/performed experiments in terms of mechanisms of functioning. However, we observed that some of them found complex situations (i.e., situations involving interaction between solid and liquid particles) harder to explain than simpler ones. This was not observed in students making up part of the other group (the ones that were proposed with a traditional macroscopic approach to the study of surface phenomena). This aspect deserves more attention and will be one of the topics of further study of the collected data.

## 6. Discussion and Conclusions

The results reported here show that a mesoscopic modelling approach can promote an improvement in thinking in terms of mechanisms of functioning (i.e., particle interaction as the cause of surface phenomena).

The computer simulations proposed to the students during the TLS activities allowed them to observe the physical systems at equilibrium and also their evolution over time. For instance, when the liquid-liquid interaction is comparable to the solid-liquid one, the students were able to observe a concave meniscus (see Figure 2). When the liquid-liquid interaction is greater that the solid-liquid one, the students observed a convex meniscus (see Figure 3).

Moreover, when the solid-liquid force is slightly bigger than the liquid-liquid one, the students were able to observe a convex meniscus and the rise of the liquid inside the capillary [18]. The students were able to easily visualize the principal forces applied to the mesoscopic particles of liquid and to see in real-time the effects of these forces on modifications of relevant simulation parameters. Quantitatively, the modelling approach presented here also allowed the students to simulate the behaviour of different liquids by setting the fundamental quantities on which the behaviour itself depends (for instance, density, viscosity and intensity of the interaction forces). As an output of the computer simulation, students were able to obtain the value of the surface tension in a given configuration or verify the Young-Laplace law (see Figure 5).

The simulation program also allowed the students to capture short movies of the simulations. These could be discussed later with peers and with the teachers, fostering students' discussion and debating skills, and allowing them to reflect and build on special aspects that were possibly not highlighted previously.

Research has shown [5] that the use of a modelling approach in science education may be a useful tool for both high school and undergraduate students in many scientific fields. Our results show that allowing students (and teachers) to control relevant physical parameters in a simulation can enhance the understanding of the ways in which knowledge is constructed and can improve students' reasoning. Particularly, it can improve the understanding of the mechanism of functioning at the basis of phenomena that are observed or studied. Of course, further studies are necessary to better correlate the typical quantities of the model with the macroscopic ones in the specific case of surface phenomena, so as to allow students to also explain complex situations related to real-life situations that are particularly relevant for them.

**Author Contributions:** Conceptualization, C.F. and O.R.B.; methodology, C.F. and O.R.B.; software, O.R.B.; validation, G.T.; formal analysis, C.F. and G.T.; investigation, G.T.; resources, C.F., A.A.G., O.R.B. and G.T.; data curation, A.A.G. and G.T., writing—original draft preparation, C.F. and O.R.B.; writing—review and editing, C.F., A.A.G., O.R.B. and G.T; visualization, A.A.G.; supervision, C.F.; project administration, C.F.; funding acquisition, C.F. All authors have read and agreed to the published version of the manuscript.

**Funding:** This research received no external funding.

**Institutional Review Board Statement:** Not applicable.

**Informed Consent Statement:** Informed consent was obtained from all subjects involved in the study.

**Data Availability Statement:** The data presented in this study are available on request from the corresponding author.

**Conflicts of Interest:** The authors declare no conflict of interest.

## Appendix A

### Questionnaire and student typical responses

Question 1: The liquid inside a glass capillary tube has a concave meniscus (see the figure). Why? Explain in terms of forces.

(1) The student explains in terms of equilibrium of the adhesive and cohesive forces in the liquid
(2) The student describes in terms of cohesive forces and adhesive forces between liquid and glass. The student does not go into detail
(3) The student explains in terms of adhesive forces between liquid and glass
(4) The student describes in terms of forces acting among liquid molecules
(5) The student explains in terms of inter-molecular forces. The student introduces the adhesive forces between liquid and the walls of the capillary tube
(6) The student explains in terms of adhesive forces between liquid and solid and cohesive forces of the liquid
(7) The student describes the observed phenomenon by introducing the surface tension concept in a general sense
(8) The student explains in terms of surface tension of the liquid
(9) No response

Question 2: A liquid does not wet the glass it is in contact with. Why? Explain in terms of forces.

(1) The student describes in terms of general properties of the molecules
(2) The student describes in terms of cohesive forces and adhesive forces with the glass. The student does not go into detail
(3) The student explains in terms of adhesive force with the glass

(4) The student describes in terms of properties of material
(5) The student describes in terms of liquid inter-molecular forces
(6) The student explains in terms of inter-molecular forces
(7) The student explains in terms of adhesive forces between liquid and solid and cohesive forces of the liquid
(8) The student explains in terms of surface tension of the liquid
(9) The student describes in terms of general forces (repulsive forces, capillary forces, etc.)
(10) No response

Question 3: Consider a toy boat floating on the surface of the water contained in a tank. After dropping a few drops of soap in the water with a dropper, it is observed that the boat starts to move. Why? Explain in terms of the forces acting on the boat.

(1) The student explains in terms of physical-chemical properties of the soap
(2) The student describes in terms of interaction forces between water and soap
(3) The student describes in terms of general forces (elastic force for example)
(4) The student explains in terms of formation/breaking of inter-molecular bonds
(5) The student explains the phenomenon indicating the surfactant as the agent that cause surface tension breaking
(6) The student describes in terms of surfactant properties
(7) The student explains the cause of the boat's motion with a change in the structure and/or bonds of the liquid
(8) The student introduces (describes) surface tension as the cause of the boat's motion
(9) No response

Question 4: What are the units of measurement of surface tension? How is it possible to obtain them?

(1) The student indicates the units but does not explain how to derive them
(2) The student indicates the units and explains how to derive them
(3) The student indicates the units and derives them through a mathematical relation
(4) The student indicates the physical quantities but not the relationship between them (units)
(5) The student indicates the physical quantities and explains how to derive them
(6) No response

Question 5: The number of water droplets required to completely cover the surface of a coin is greater than the number of seed oil droplets required to cover the surface of a coin identical to the first one. Explain what this phenomenon might be related to.

(1) The student describes in terms of liquid properties (density, surface tension etc.)
(2) The student describes in terms of adhesive and/or cohesive forces in the liquid
(3) The student explains in terms of adhesive and/or cohesive forces in the liquid
(4) The student describes macroscopically
(5) The student explains in terms of surface tension of the liquid
(6) The student explains in terms of liquid inter-molecular interactions
(7) The student describes in terms of surface properties
(8) No response

Question 6: Consider three capillary tubes of the same material and of the same diameter each dipped in three tanks containing water, mercury and oil, respectively. Represent how each liquid will be arranged inside the capillary tubes graphically. Explain in terms of forces.

(1) The student describes in terms of liquid properties (density, capillarity, surface tension etc.)
(2) The student describes in terms of general forces
(3) The student describes in terms of adhesive and/or cohesive forces in the liquid
(4) The student explains in terms of adhesive and/or cohesive forces in the liquid
(5) The student describes in terms of liquid inter-molecular interactions
(6) The student explains in terms of liquid inter-molecular interactions
(7) No response

Question 7: What do you think are the "adhesive forces" and "cohesive forces"? Give some examples of contexts in which these forces are present.

(1)     The student explains among whom the forces act. The student does not give examples
(2)     The student explains among whom the forces act and gives examples
(3)     The student explains in terms of inter-molecular bonds
(4)     The student gives general examples
(5)     No response

Question 8: What difference do you think there is between these two types of forces?

(1)     The student explains in terms of the different nature of the molecules
(2)     The student explains in terms of the different nature of the substances
(3)     The student explains introducing the vectorial nature of the forces
(4)     The student explains among whom the forces act
(5)     No response

Question 9: Which quantities influence the rise of a liquid inside a capillary tube? Explain.

(1)     The student describes in terms of forces (tension, adhesive, cohesive forces etc.)
(2)     The student explains in terms of forces (tension, adhesive, cohesive forces etc.)
(3)     The student describes in terms on general geometric properties (diameter, density, friction, pressure, length of the tube, gravity, viscosity etc.)
(4)     No response

Question 10: Do you think the soap modifies water properties? If yes, which ones?

(1)     The student describes in terms of physical-chemical properties
(2)     The student explains in terms of inter-molecular interaction
(3)     The student describes in terms of surface tension
(4)     The student explains in terms of surface tension
(5)     The student explains in terms of adhesive/cohesive forces (surfactants)
(6)     The student describes in terms of general physical quantities and/or forces (density, surface tension etc.)
(7)     The student cites everyday/common life situations
(8)     No response

Question 11: If you put some soap into water and deposit a drop of this mixture on a horizontal plane, you will notice that the water is evenly distributed on the surface not forming a real drop. Explain this phenomenon.

(1)     The student describes in terms of physical-chemical properties
(2)     The student explains in terms of inter-molecular interaction
(3)     The student describes in terms of surface tension
(4)     The student explains in terms of surface tension
(5)     The student explains in terms of adhesive/cohesive forces (surfactants)
(6)     The student describes in terms of general physical quantities and/or forces (density, surface tension etc.)
(7)     The student describes on the basis of observed phenomena
(8)     No response

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
