# Peer review of "Outcomes of a Teaching Learning Sequence on Modelling Surface Phenomena in Liquids"

_education, doi:10.3390/educsci13040425_

Round 1

Reviewer 1 Report

I am not sure if this paper is about how to represent a mesoscopic modelling of liquid or epistemological profiles of the students in learning physics. For a journal entitled Education Sciences, readers are expected to read education sciences such as epistemological profiles of students in learning physics rather than going through the mathematical modelling of a physical phenomenon. I would expect to read how computer simulation in learning physics can develop a more sophisticated epistemological profile among the students. Hence, I would expect the literature review related to epistemological beliefs of students in learning physics and also the implication of using computer simulations in developing sophisticated epistemology. 

Author Response

We thank the referee for the interesting and constructive reviews and for the comments. They allowed us to greatly improve our manuscript.

Comment:

I am not sure if this paper is about how to represent a mesoscopic modelling of liquid or epistemological profiles of the students in learning physics. For a journal entitled Education Sciences, readers are expected to read education sciences such as epistemological profiles of students in learning physics rather than going through the mathematical modelling of a physical phenomenon. I would expect to read how computer simulation in learning physics can develop a more sophisticated epistemological profile among the students. Hence, I would expect the literature review related to epistemological beliefs of students in learning physics and also the implication of using computer simulations in developing sophisticated epistemology.

Our answer:

We understand the referee’s comment and modified the manuscript in the sense suggested by him/her. We shortened Section 2 and 3, and improved the discussion on how the use of computer simulation in learning physics can improve the development of reasoning lines more oriented to explanation, rather than to description. We also added some more qualitative data that support our conclusions. Finally, we added references about student epistemological beliefs about models and their use in learning science.

Author Response

We thank the referee for the interesting and constructive review and for the comments. They allowed us to greatly improve our manuscript.

Comment:

“Page 2, line 93. Authors speak about “the pressure of the generic particle I”. Pressure is an emergent mesoscopic (at least; possibly macroscopic) quantity, not appropriate for a single particle. I suggest clarifying the meaning of “pressure” for a single (mesoscopic, in this case) particle.

Our answer:

Thank you for the useful comment. We added detail on the meaning of “pressure” for a single (mesoscopic, in this case) particle.

Comment:

“Moreover, at the same line, “I” should be lower case “i”.”

Our answer:

Fixed.

Comment:

“Page 3, Figure 1. The Fij-vs-rij relation for the interparticle interaction suggests a millimetric order of magnitude for the “particle” dimension. The same order of magnitude (or, maybe, tenths mms) is explicitly shown in Figures 2, 3, 4. Such a dimension appears to be quite enormous on the molecular (or even super- molecular) scale and can be pedagogically misleading. So, I think some explanation is in order regarding the physical meaning of the assumed dimension for meso-scale particles.”

Our answer:

We added a sentence to better clarify the nature of the mesoscopic particles in our model.

Comment:

“Captions of Figure 2 and Figure 3. Maybe “Black dashed ROWS” should be “Black dashed ARROWS”? “

Our answer:

Fixed.

Comment:

“Page 6, lines 203 and 208. The concept of “virial radius” is familiar (as I know) only for astrophysicists in the context of gravitational clustering of galaxies. Given the possible readership of the present paper, I think some explanation (e.g. in a footnote) or some pedagogical-grade bibliographic reference for this concept is needed.

Our answer:

Thank you for the useful comment. We added some sentences to better explain the meaning of the Virial Theorem and the virial radius in our case.

Comment:

“Page 6, Equation 5. It’s worth specifying in the summation symbol ?? ≠ ??.”

Our answer:

We agree with the reviewer. We modified Eq. 5 according to this comment.

Comment:

“Page 6, Figure 5 and line 220. Authors use Equation (4) to fit data in Figure 5. This is meaningful in order to highlight the linear relation between the computational “experimental” data PT and the inverse radius. However, given the arbitrary units for PT, I think that the specific value obtained for the "surface tension" is devoid of physical meaning and therefore it is not appropriate to report it (moreover with 5 significant figures).”

Our answer:

We agree with the reviewer. We removed the surface tension value.

Reviewer 3 Report

Dear Authors

Congratulations on the article proposed for publication. The article complies with the editorial standards of the journal.

However, some corrections are required

Article review - Outcomes of a Teaching Learning Sequence on modelling sur- 2 face phenomena in liquids.

·         The title of paper has 14 words.

·         Authors respect the structure of article template, and the Authors Guidelines. The article is original research manuscripts

The paper is following chapter.

1.      Introduction

The introduction chapter has briefly placed the study in a broad context and highlight why it is important. It defines the purpose of the work and its significance.

2.     The Model

The materials and methods are described to allow others to replicate and build on published results.

3.     Some Examples of Simulation Activities 150 3.1.

3.1.  Menisci Formation

3.2. The pressure inside a droplet

4.     The student sample and the Teaching/Learning Sequence

4.1.    The student sample

4.2. The TLS

5.     The results

6.     Discussion and conclusions

Observations

For the computer simulation activities aren't give the name and version of any software used. (In this paper we discuss the effect of modelling and computer simulation activities”)

The bibliography is not in alphabetical order

Education Sciences | Instructions for Authors (mdpi.com)

Author Response

We thank the referee for the interesting and constructive review and for the comments. They allowed us to greatly improve our manuscript.

Comment:

For the computer simulation activities aren't give the name and version of any software used. ( „In this paper we discuss the effect of modelling and computer simulation activities”)

Our answer:

We did not use any commercial software, but rather coded a custom one, that allows students and teachers to directly modify the simulation parameters by means of a simple text file. We used MathLab for graphs and movies.

We also described how the simulation is used by students/teachers.

Comment:

The bibliography is not in alphabetical order

Our answer:

We used the Word template for submission and followed the instruction reported at Education Sciences | Instructions for Authors (mdpi.com).

Particularly, we followed the text: “References: References must be numbered in order of appearance in the text (including table captions and figure legends) and listed individually at the end of the manuscript.”

Reviewer 4 Report

This manuscript merges the presentation of a fluid dynamics simulation tool: the Smoothed-Particle Hydrodynamics (SPH), and the analysis of a teaching-learning sequence (TLS) that involved using SPH for improving the learning of interfacial phenomena in fluid dynamics. Its main strength and novelty are to propose a systematic use of SPH for learning fluid dynamics.  

However, I noticed several drawbacks in the manuscript that I can describe as follows:

(1) Manuscript structure: There needs to be more integration between the paired sections 2-3 and 4-5, the former being too focused on SPH and fluid dynamics and the latter on the teaching-learning sequence (TLS). For example, thinking about using SPH in classroom environments, it could be relevant to give more details about how to implement SPH (for instance, if you use DualSPHysics, PreonLab, etc., and in which kind of computers) in order to invite other teachers to implement the methodology. Conversely, I missed details about how the software was used (or more about the activities) to improve the questionnaire results. 

In the present version, the manuscript merges two different papers without enough bridges between them. 

(2) Discussion and conclusions: Some sentences in the section need to be stronger or better justified by the data presented in the manuscript (phrases like "seems to promote"; "We think" give an unclear message). Also, I do not see enough support for the sentence: "For instance, when the solid-liquid force is slightly bigger than the liquid-liquid one, we can observe a convex meniscus and the rise of the liquid inside the capillary. At least, I do not see it as supported by the presented results.

(3) Fluid dynamics (Introduction and Sections 3-4). 

I agree with the interest in using SPH to study fluid dynamics and the need for careful consideration of basic interfacial phenomena like drops and meniscus [18, 19, 20]. 

However, those interfacial phenomena are easily visualized by experiments (see, for instance, the beautiful book "Capillarity and Wetting Phenomena: Drops, Bubbles, Pearls, Waves" by P. G. de Gennes, F. Brochard-Wyart, and D. Quéré). Therefore, there is a need for a better justification for using numerical simulations in pedagogical contexts. In particular, some of the model's variables have a more complex interpretation than what can be observed (or measured) experimentally. 

Specific questions: 

(3a) What is the physical meaning of H? ("the smoothing length")

Can its value be included in figure 1?

(3b) Similar question for the artificial viscosity. How is it set in practice? Is it related to \nu?

(3c) In section 3: Is the tank for studying menisci formation 2D or 3D? 

(3d) What is the "capillary length" for this system? I ask this because one expects the capillary length to set the characteristic length of the menisci.

(3e) (curiosity-driven question) As there is no gravity in the study of drops. Is it possible to observe drop oscillations? There are nice studies about their dependence on drop size (see, for instance, the classic JFM1991 "Experimental and theoretical investigation of large-amplitude oscillations of liquid droplets" by Becker, Hiller & Kowalewski).

(3f) I am concerned about using "arbitrary units" at some places in this section because they made interpretation and comparisons harder. Particularly in the fitted value for the surface tension, I cannot compare it with the typical value from the water-air interface involved. 

(4) Teaching/Learning Sequence (sections 4-5 and Appendix)

(4a) Through section 4, there are several references to experimental activities (lines 231; 270; 306). Could you describe them in more detail?

(4b) While it is the topic for another report, I would like to see more details about how the results go from the answers to the questionnaire to the classification between the three categories presented in table 1. 

(4c) I am concerned about the questionnaire's strong focus on forces. While I agree with the interpretation in terms of forces, interfacial phenomena could also be understood in terms of surface energy. Therefore, by restricting ourselves to forces, we also restrict the learning of the phenomena. 

In conclusion, although the goal and the approach of the manuscript may be helpful for teaching-learning fluid dynamics, I have significant concerns about the manuscript in the present form. Consequently, I cannot recommend its publication.

Author Response

We thank the referee for the interesting and constructive review and for the comments. They allowed us to greatly improve our manuscript.

Comment:

(1) Manuscript structure: There needs to be more integration between the paired sections 2-3 and 4-5, the former being too focused on SPH and fluid dynamics and the latter on the teaching-learning sequence (TLS).

Our answer:

We understand the referee’s comment and modified the manuscript in the sense suggested by him/her. We shortened Sections 2 and 3, and improved the discussion on how the use of computer simulation in learning physics can improve the development of reasoning lines more oriented to explanation, rather than to description. We also added some more qualitative data that support our conclusions. Finally, we added references about student epistemological beliefs about models and their use in learning science.

Comment:

“For example, thinking about using SPH in classroom environments, it could be relevant to give more details about how to implement SPH (for instance, if you use DualSPHysics, PreonLab, etc., and in which kind of computers) in order to invite other teachers to implement the methodology. Conversely, I missed details about how the software was used (or more about the activities) to improve the questionnaire results.”

Our answer:

We added more detail on how we implemented the SPH method, made the code and how the simulation can be used by students/teachers.

Comment:

(2) Discussion and conclusions: Some sentences in the section need to be stronger or better justified by the data presented in the manuscript (phrases like "seems to promote"; "We think" give an unclear message). Also, I do not see enough support for the sentence: "For instance, when the solid-liquid force is slightly bigger than the liquid-liquid one, we can observe a convex meniscus and the rise of the liquid inside the capillary. At least, I do not see it as supported by the presented results.

Our answer:

We modified the manuscript by following the referee’s comment and removed the terms that can foster unclear messages. Also, the text "For instance, when the solid-liquid force is slightly bigger than the liquid-liquid one, we can observe a convex meniscus and the rise of the liquid inside the capillary” was modified relating it to the use students did of the simulation and to the advantages they could obtain from that use. More generally, all section 6. Was enhanced by following the referee’s nice suggestions.

Comment:

“However, those interfacial phenomena are easily visualized by experiments (see, for instance, the beautiful book "Capillarity and Wetting Phenomena: Drops, Bubbles, Pearls, Waves" by P. G. de Gennes, F. Brochard-Wyart, and D. Quéré). Therefore, there is a need for a better justification for using numerical simulations in pedagogical contexts. In particular, some of the model's variables have a more complex interpretation than what can be observed (or measured) experimentally.”

 Our answer:

According to our aims, the simulation is not only useful to easily visualize a phenomenon that would be difficult to observe experimentally. We have added details on the possibility of modifying some simulation parameters to better understand the effect of these modifications on the behaviour of the liquid.

Comment:

“What is the physical meaning of H? ("the smoothing length")”

 Our answer:

We added more detail about the physical meaning of the quantity H.

Comment:

“Can its value be included in figure 1?”

 Our answer:

It could be included. However, for clarity sakes we preferred representing only the interaction force as a function of the interparticle distance.

Comment:

“Similar question for the artificial viscosity. How is it set in practice? Is it related to \nu?”

 Our answer:

We have greatly simplified the description of the model especially by removing mathematical details of the SPH methodology, also in agreement with comments by other reviewers. For this reason, many quantities such as the artificial viscosity are now no longer reported in the manuscript.

Comment:

“In section 3: Is the tank for studying menisci formation 2D or 3D?”

 Our answer:

The tank is 2D, due to the computational load required by a 3D one

Comment:

“What is the "capillary length" for this system? I ask this because one expects the capillary length to set the characteristic length of the menisci.”

 Our answer:

The simulation of menisci formation is currently for qualitative purposes only. For this reason, we have not reported any values of typical quantities such as the "capillary length".

Comment:

“(curiosity-driven question) As there is no gravity in the study of drops. Is it possible to observe drop oscillations? There are nice studies about their dependence on drop size (see, for instance, the classic JFM1991 "Experimental and theoretical investigation of large-amplitude oscillations of liquid droplets" by Becker, Hiller & Kowalewski).”

 Our answer:

Yes, it is possible. Many thanks for the reference. We will study it to see if suggestions can be obtained for further research.

Comment:

“I am concerned about using "arbitrary units" at some places in this section because they made interpretation and comparisons harder. Particularly in the fitted value for the surface tension, I cannot compare it with the typical value from the water-air interface involved.

 Our answer:

We agree with the reviewer. We removed the surface tension value obtained by linear fitting.

Comment:

(4a) Through section 4, there are several references to experimental activities (lines 231; 270; 306). Could you describe them in more detail?

 Our answer:

We added detail to some of the experimental activities, as requested.

Comment:

(4b) While it is the topic for another report, I would like to see more details about how the results go from the answers to the questionnaire to the classification between the three categories presented in table 1.

 Our answer:

Thanks for this comment. We added a table (Table 2) giving more detail on the process to go from the answers to the questionnaire to the classification between the three categories presented in table 1.

Comment:

(4c) I am concerned about the questionnaire's strong focus on forces. While I agree with the interpretation in terms of forces, interfacial phenomena could also be understood in terms of surface energy. Therefore, by restricting ourselves to forces, we also restrict the learning of the phenomena.

Our answer:

We chose to focus the questionnaire (and all the TLS activities) on forces because the students were at secondary level. An approach based on energy (a thermodynamical approach) can appear abstract to those students, also because it is unrelated to what students commonly study in school with respect to surface phenomena. So, we chose to concentrate on the forces, also because they can be easily visualized in the simulations.

Round 2

Reviewer 1 Report

There is no literature presented in the paper on the previous research in scientific epistemology and TLS in this paper, hence the discussion was poorly done because it has not discussion to contribution to new knowledge in either scientific epistemology or TLS. Moreoever, this is an educational journal, hence there is no need to write extensively the physics modelling and equations. It would be appropriate to do so in science journals.

Author Response

Thank you again for your comments.

We answer to the issues you pose in the following:

COMMENT: There is no literature presented in the paper on the previous research in scientific epistemology and TLS in this paper, hence the discussion was poorly done because it has not discussion to contribution to new knowledge in either scientific epistemology or TLS.

OUR ANSWER: We added some literature references on TLS and their use in Physics Education Research. Regarding the refeences you request on scientific epistemology, we remark that the main focus of the manuscript is not on that topic, so we do not added any more about that

COMMENT: Moreoever, this is an educational journal, hence there is no need to write extensively the physics modelling and equations. It would be appropriate to do so in science journals.

OUR ANSWER: We understand your issue. However, this is a paper to be published in a special issue on Innovation in Teaching/Learning Physics, so we think that detail on physics modelling and equations is needed and useful for the reader, that is probably also a specialist in physics (or science) education. 

Reviewer 4 Report

I thank the authors for taking into account the referees' suggestions. The manuscript has greatly improved. However, some of the concerns I presented in my previous report were solved only partially. 

My current main concern is about the final part: Discussion and Conclusions. This section mentions activities that seem valuable but are not covered in the manuscript (lines 437-438). Also, it describes outcomes from activities (445-454) not documented in the manuscript. In that sense, conclusions are not well supported.

Related to the previous comment (and also mentioned in my previous report), while improved, I still see the manuscript as two halves, not having enough conversation between them. For instance, the descriptions of activities in the TLS section are brief, while the detailed numerical results from section 3 are presented without mentioning any connection to the TLS.  

A final concern comes from the fluid mechanics' standpoint. Almost every surface phenomenon in liquids involves the capillary length as a fundamental length scale (probably the only exception is zero-gravity environments). Using other length scales without comparing them to the capillary length makes that part incomplete. 

Author Response

Thank you again for your comments, once more very useful for us to improve our paper.

We answer to the issues you pose in the following:

COMMENT: My current main concern is about the final part: Discussion and Conclusions. This section mentions activities that seem valuable but are not covered in the manuscript (lines 437-438). Also, it describes outcomes from activities (445-454) not documented in the manuscript. In that sense, conclusions are not well supported.

OUR ANSWER: We took this comment in great consideration and restructured the text covering the activities that were not well discussed, or citing literature references were the reader can find more detail (due to the space constraints in this manuscript, that does not allow us to be more explicit)

COMMENT: Related to the previous comment (and also mentioned in my previous report), while improved, I still see the manuscript as two halves, not having enough conversation between them. For instance, the descriptions of activities in the TLS section are brief, while the detailed numerical results from section 3 are presented without mentioning any connection to the TLS.

OUR ANSWER: We restructured  Section 4.2, describing the TLS in more detail and more ordinately, and relating more strongly the description of the modelling activities with the content presented in Section 3

  COMMENT: A final concern comes from the fluid mechanics' standpoint. Almost every surface phenomenon in liquids involves the capillary length as a fundamental length scale (probably the only exception is zero-gravity environments). Using other length scales without comparing them to the capillary length makes that part incomplete.

OUR ANSWER: Thank you for this very interesting comment. We added detail on the "capillary lenght" issue you mentioned in Section 4.2, with respect to the activities that students could do when simulating the rise of a liquid in a capillary tube.